# Selected Simple Natural Antimicrobial Terpenoids as Additives to Control Biodegradation of Polyhydroxy Butyrate

**DOI:** 10.3390/ijms232214079

**Published:** 2022-11-15

**Authors:** Ahmad Fayyazbakhsh, Marek Koutný, Alena Kalendová, Dagmar Šašinková, Markéta Julinová, Markéta Kadlečková

**Affiliations:** 1Department of Environmental Protection Engineering, Faculty of Technology, Tomas Bata University in Zlín, T. G. Masaryk Square 5555, 76001 Zlín, Czech Republic; 2Department of Polymer Engineering, Faculty of Technology, Tomas Bata University in Zlín, T. G. Masaryk Square 5555, 76001 Zlín, Czech Republic; 3Centre of Polymer Systems, University Institute, Tomas Bata University in Zlín, Tr. T. Bati 5678, 76001 Zlín, Czech Republic

**Keywords:** polyhydroxybutyrate (PHB), biodegradation, essential oils (EOs), mechanical properties

## Abstract

In this experimental research, different types of essential oils (EOs) were blended with polyhydroxybutyrate (PHB) to study the influence of these additives on PHB degradation. The blends were developed by incorporating three terpenoids at two concentrations (1 and 3%). The mineralization rate obtained from CO_2_ released from each sample was the factor that defined biodegradation. Furthermore, scanning electron microscope (SEM), differential scanning calorimetry (DSC), and dynamic mechanical analysis (DMA) were used in this research. The biodegradation percentages of PHB blended with 3% of eucalyptol, limonene, and thymol after 226 days were reached 66.4%, 73.3%, and 76.9%, respectively, while the rate for pure PHB was 100% after 198 days, and SEM images proved these results. Mechanical analysis of the samples showed that eucalyptol had the highest resistance level, even before the burial test. The other additives showed excellent mechanical properties although they had less mechanical strength than pure PHB after extrusion. The samples’ mechanical properties improved due to their crystallinity and decreased glass transition temperature (Tg). DSC results showed that blending terpenoids caused a reduction in Tg, which is evident in the DMA results, and a negligible reduction in melting point (Tm).

## 1. Introduction

Plastics have been used indiscriminately for decades, but they have eventually raised strong ecological concerns. In this regard, the negative influence of petroleum-based plastics on the environment has led to the development of biobased and biodegradable plastics [1,2]. Plastics derived from renewable resources have received significant attention as an environmentally friendlier alternative to petroleum-based polymers due to their biodegradability, which can occur in aerobic, and sometimes even anaerobic, conditions [3,4,5,6]. One of the most studied materials in this class is polyhydroxybutyrate (PHB), which can be used in many applications. It is a thermoplastic polymer that is renewable as it is produced by many Gram-negative and Gram-positive bacteria [7] as carbon- and energy-storage material [8,9].

The chemical structures of polymers, as well as the environment to which they are subjected, have a major impact on how quickly they degrade. Other factors that have a significant impact on biodegradability include temperature, oxygen levels, humidity, the predominance of particular microbial communities, and the availability of nutrients. PHB is known as a completely degradable polymer in a variety of controlled and natural biologically active conditions [10]. PHB is a thoroughly studied biodegradable polymer with biocompatibility and biodegradability, and its properties are suitable in food-packaging and agricultural applications; yet, due to characteristics, such as its melting temperature, lack of flexibility, and low extension to break, its commercial applicability has been limited so far [11]. Moreover, it has a high degree of crystallinity, as well as low impact strength, as compared to other biodegradable polymers [12].

Despite having some favorable properties, PHB has some limitations. First, this polymer’s price and availability are the critical factors restricting its broader use so far. Second, the material has less favorable mechanical and thermal properties [13,14]. To eliminate these problems, many studies have been performed [9,15], using various types of copolymers (which could provide materials with more favorable morphology) [16,17], crosslinking agents [18], natural compounds [19,20], antimicrobial agents [21], and chain extenders [22,23]. Third, the rapid biodegradation of PHB is a drawback that restricts its use in some agricultural applications, especially those which require direct contact with soil.

The latter issue could be controlled by adding compounds with mild antimicrobial properties that can slow down, but not completely prevent, the biodegradation of the material in the soil. Natural, simple terpenoids appear to be attractive candidates for such compounds. Previously, terpenoids were studied in PHB materials mainly to prevent the adsorption of pathogenic bacteria on its surface. Narayanan and his colleagues [24] studied the antimicrobial effect of the addition of eugenol to PHB against Salmonella typhimurium, Staphylococcus aureus, and Escherichia coli. The results showed that eugenol–PHB films, in conjunction with another antimicrobial, pediocin, could effectively control the growth of food spoilage microorganisms, food-contaminating molds, and foodborne pathogens. Oliveira et al. [25] studied the antibacterial effect of eucalyptol and carvacrol against Pseudomonas fluorescens, Aeromonas hydrophila, and Listeria monocytogenes, as model organisms in a mixed culture. They reported that carvacrol’s minimum inhibition concentration (MIC), which shows the minimum required concentration of an antimicrobial [26], was about 30 times lower than eucalyptol, but both had sufficient potential as antimicrobial agents. Porfirio et al. [27] and Huang et al. [28] studied the antibacterial and antifungal effects of carvone and citral. The first study reported that the MIC of carvone was 4 times higher than that of citral; however, Huang and his research group found that, as an antifungal, the MIC of citral was about 67% higher than that of carvone.

The compounds in the discussed group can also act as plasticizers in polymeric materials in somewhat higher concentrations [29,30]. Mangeon et al. [4] studied the influence of three different terpenoids as plasticizers on PHB. Their results revealed that these additives enhanced the elongation at break by up to 650%. Moreover, a reduction in Tg and the storage modulus was observed after adding these compounds.

As explained, previous studies did not investigate the influence of antibacterial terpenoid addition on the biodegradation of PHB-based materials in soil. Here, we explore this phenomenon and show a way to control the biodegradation of such materials. These findings can be further exploited to develop more functionally advanced materials to be used in agriculture, e.g., mulching films or other products that should be completely biodegradable in the soil, but which must retain their properties during their limited service lives.

## 2. Results and Discussion

### 2.1. Preliminary Biodegradation Experiment

As a preliminary study, 10 PHB materials with terpenoids were prepared to find the most effective ones. The majority of the compounds were selected based on their MICs. The materials were evaluated after 65 days of mineralization and a simple burial test in which the disintegration of the samples was observed. The results indicated that thymol, eucalyptol, and limonene were the most promising additives for slowing the biodegradation of the materials in the soil, while the effect of the other tested compounds seemed marginal.

### 2.2. Mineralization of Materials in Soil

The CO_2_ evolution measurement monitored the mineralization of the materials. The data proved a substantial retardation of biodegradation by all of the additives (Figure 1). As can be seen, the blend with 3% eucalyptol showed the best resistance against degradation. Although pure PHB had been degraded completely prior to 200 days, PHB blended with 3% limonene was the second-best sample in retarding biodegradation, and 3% thymol showed moderate retardation of biodegradation. In this regard, after 226 days of incubation, 3% eucalyptol, 3% thymol, and 3% limonene reached 66.4, 77, and 73.3% of mineralization, respectively, due to their antimicrobial activities [19,31]. The influence of limonene in retarding biodegradation was more substantial than that of thymol due to its higher antifungal activity [32]. The film degradation in the preliminary study proved the more substantial influence of eucalyptol in PHB degradation, which was also due to its high antimicrobial and antioxidant activities [33]. Furthermore, blending essential oils with PHB is a technique for enhancing the polymer’s crystallinity, which reduces biodegradation [19].

### 2.3. Visual and Microscopic Analysis

The physical shapes of the samples are shown in Figure 2. As can be seen, the blend with 3% of eucalyptol was the only sample that could be identified after 32 days; the other samples had disintegrated prior to that. The images also depict that samples with thymol and limonene were more resistant than pure PHB.

The biodegradation results were supported qualitatively by SEM images (Figure 3). These images show that pure PHB has a higher potential to degrade than that of the other samples, as is evident from its substantial biodeterioration. The material with 3% eucalyptol showed a relatively smooth surface, which could mean a higher level of degradation resistance than that of the other samples. Moreover, the sample with 3% thymol showed better resistance to degradation than did the sample with limonene and pure PHB, as depicted in Figure 2.

Fungi hyphae are easily discernible in the pictures, supporting the assumption that fungi play an essential role in PHB biodegradation in soil [34,35,36]. The materials, however, were not covered by a dense biofilm of cells. The very distinct structure of the crystalline lamellae revealed during the biodegradation is also clearly visible and suggests that the biodegradation of highly crystalline PHB starts in the more amorphous matter in the center of the crystallites and between the lamellae.

### 2.4. DSC Analysis

Thermal parameters, such as melting point, glass transition temperature, and they crystallinity of the PHB samples were studied using differential scanning calorimetry (DSC). The obtained data are summarized in Table 1 and Table 2. As can be seen, the addition of the selected compounds caused a Tg reduction (Table 3). The decrease in Tg values showed that eucalyptol, thymol, and limonene were miscible with the amorphous phase of PHB and that they increased the molecular mobility of the polymer matrix [3]. The biggest Tg changes were observed in materials with limonene (29%). Furthermore, the Tm was evaluated. After the addition of the selected compounds, two melting transitions (Tm1 and Tm2) were observed in the melting region of the PHB blends, while pure PHB showed only one sharp maximum at 172 °C (Tm2). After partial biodegradation, two melting transition temperatures were observed in all of the tested compositions. This indicates that the prepared mixtures underwent the melting and recrystallization of their subphases. During biodegradation, only minor changes in the Tm were marked in the PHB modified by the terpenoids. Furthermore, all mixtures displayed a very weak degree of cold crystallization Tcc (Table 2). It was found that the Tcc moves slightly towards higher values (max. change of 3%) with increased biodegradation time. It seems that Tcc is affected significantly by neither terpenoid presence nor degradation time.

The crystallinity of the materials during their biodegradation was also studied with the help of DSC (Table 3). As shown in Figure 4, microorganisms started to degrade the materials from the amorphous part, so the crystalline parts were exposed. The DSC data indeed support this assumption (Table 1 and Table 2). As can be seen, pure PHB was substantially degraded after 16 days (Figure 3), which was faster than that of other samples. The drop in the crystallinity of the PHB at different degradation times was faster than in the PHB/terpenoid blends (Table 3), except for the composition with eucalyptol. At the beginning of the experiment and after 4 days of degradation, the crystallinity of the tested samples was similar (a max. change of 5% compared to pure PHB), but after 8 days, the samples blended with terpenoids exhibited higher values of crystallinity compared to the pure PHB samples (Table 3, Figure 5). In the case of the PHB/eucalyptol blend, a rapid decrease in crystallinity had already been observed after 4 days, but a renewed increase was also detected in the sample after 32 days. This phenomenon was asserted by Beltrami and her colleagues [36], who proposed that enzymatic degradation started first in the amorphous phase, thus increasing the crystallinity of the samples. The PHB/limonene mixture showed the most stable level of crystallinity over time. The explanation of this behavior may also be connected to the antimicrobial properties of terpenoids.

### 2.5. Mechanical Properties

We used DMA to investigate the dynamic mechanical properties of the materials during biodegradation. Figure 6 presents the complex storage modulus E* trend for PHB compositions with terpenoids. In general, the E* remained constant and high in the glassy state and then dropped dramatically following the Tg before stabilizing in the rubbery state. The pure PHB and PHB with 3% limonene exhibited the fastest drop from a glassy to a rubbery state at the beginning of the experiment (0 days). Other compositions showed a more gradual decline during the thermal-mechanical exposition on day 0.

The E* values for 30 °C are summarized in Table 3 and Figure 5. The complex modulus (Figure 6) decreased after adding terpenoids. The most significant drop was observed in the samples containing limonene. These data are in correlation with the DSC measurements. Terpenoids act like plasticizers in the PHB matrix and could decrease intermolecular stresses throughout the polymer chain, enhancing chain mobility and flexibility [30]. During incubation in the soil, another decrease occurred. After 8 days, the pure PHB showed shallow values for E*, close to 0 MPa. The composition with thymol also exhibited very low values for E* modulus. For this material, a further decline was recorded between 8 and 16 days of degradation. The material lost 99% of the E* modulus when compared to the same material before degradation.

On the other hand, stable E* values were exhibited by the material containing eucalyptol. This effect could relate to the antimicrobial properties of eucalyptol and the higher stability of these materials in the soil environment. During biodegradation, PHB with 3% limonene showed a decrease in the E* modulus of about 30%. The value was then stable after 8 and 16 days of incubation. Furthermore, a very small increase after 16 days was observed in this mixture. This effect could be related to the increasing degree of crystallinity, which could have enhanced the rigidity of the samples because the growing level of crystallinity led to a restriction of movement in the polymer chains [37]. Another reason for this behavior could be connected to terpenoid loss during incubation in soil.

## 3. Materials and Methods

### 3.1. Materials

The PHB used in this research was in powder form, and it was obtained from Tiatan Biologic Materials Co, Ltd. (Beilun, Ningbo, China) with a molecular weight of 66,500 g mol^−1^. The natural compounds used as additives to the PHB in this study were purchased from Sigma Aldrich and are summarized in Table 4.

### 3.2. Sample Preparation

The additives were mixed well in a bowl with pure PHB powder to reach the given concentrations before being loaded into the micro-extruder. A twin-screw micro-extruder (HAAKE Minilab, Thermo Fisher Scientific, Waltham, MA, USA) was used in this study. The processing temperature was set at 185 °C, with a speed of 50 rpm. A period of 2 min of mixing at the given conditions was evaluated as being sufficient and did not induce unwanted degradation of the material. Before the extrusion, the PHB was dried for 10 h at 100 °C. Then, compression molding was performed at the same temperature as that of the micro-extruder (185 °C) to make the films. The heating time was about 2 min, and after that, the films cooled down to room temperature for about 10 min. The specimens’ thickness was 120–130 μm, and the dimensions were 50mm × 300 mm. The same process was also used to prepare the pure PHB films used as controls.

### 3.3. Differential Scanning Calorimetry

Differential scanning calorimetry (DSC) analysis of the samples was performed on a DSC1/700 analyzer (Mettler Toledo, Columbus, OH, USA) containing a mechanical cooling system. The measurement was performed in a nitrogen atmosphere and calibrated by an indium standard. DSC was used to measure melting point, crystallinity, and Tg. The samples were placed in a DSC aluminum pan and scanned between −40 and 200 °C. First, the samples were heated from −40 to 200 °C with a heating rate of 10 °C min^−1^. This process was followed by cooling from 200 to −40 °C with a cooling rate of −10 °C min^−1^. In this method, two heating processes were introduced to the material. The initial heating scan was performed to erase the sample’s previous thermal history, and the second heating cycle of the DSC curves yielded the thermal characteristics.

### 3.4. Dynamic Mechanical Analysis

Dynamic mechanical analysis (DMA) was performed on a DMA Analyzer (Mettler Toledo, Columbus, OH, USA). The temperature ranged from −40 to 100 °C; the film-tension mode had a heating rate of 2 °C min^−1^ with a frequency of 1 Hz and a load of 0.5 N. The DMA analysis of the samples was carried out before the burial test, as well as 8 days after the burial test.

### 3.5. Biodegradation in Soil

When organic materials degrade aerobically, oxygen is consumed, and carbon is transformed to gaseous carbon dioxide (CO_2_). The proportion of solid organic carbon in the test sample, transforming into CO_2_, quantifies the mineralization. Tests were carried out in 500 mL biometric flasks. Polymer film samples were cut into 2 mm pieces (50 mg); 15 g of dry soil, a mineral medium (11 mL) [32], and 5 g of perlite were all placed in flasks (ISO 17556).

Mineralization percentage (D_t_) was calculated as:(1)$$D_{t}=\frac{[{\mathrm{CO}}_{2}{]}_{t}-[{\mathrm{CO}}_{2}{]}_{b}}{{\mathrm{ThCO}}_{2}}$$
where (CO_2_)_t_ is the accumulated CO_2_ released by each sample, (CO_2_)_b_ is the accumulated CO_2_ released by the blank flasks, and ThCO_2_ is the test material’s theoretical carbon dioxide in the test flasks. A mass spectrometer HPR-40 DSA (HIDEN Analytical, 2020, Warrington, UK) was used to assess the released carbon dioxide. For each sample, three parallel flasks were used, along with four blank flasks.

### 3.6. Scanning Electron Microscopy

Scanning electron microscopy (Phenom Pro Desktop SEM, Thermo Fisher Scientific, Waltham, MA, USA) was implemented to observe surface changes on films in high vacuum mode at an acceleration voltage of 10 kV. The samples were coated with a thin layer of Au/Pd to reduce sample damage and prevent charging.

## 4. Conclusions

This research studied the effect of blending monoterpenes into PHB. These compounds were selected from 10 terpenoids with previously proven antimicrobial properties. The preliminary study identified eucalyptol, limonene, and thymol as being the most effective in retarding the biodegradation of the PHB-based materials in the soil. The thermal and mechanical properties of the materials and their evolution during biodegradation were also studied. Eucalyptol was evaluated as being the most effective in retarding biodegradation during the biodegradation test. Limonene and thymol also provided some resistance against biodegradation (more than a 35% reduction under some conditions). The most probable reason for this was the antimicrobial activity of these compounds. Thus, it is possible to use eucalyptol, an available compound, to increase the necessary resistance of biodegradable materials based on PHB that are in contact with soil, e.g., in agricultural applications. Under the laboratory conditions supporting the maximal biodegradation rate, this material was stable for about 32 days. One might assume that under real field conditions, an even longer time could be expected. Furthermore, the combination of PHB with eucalyptol seems to offer comparable values of complex modulus with pure PHB, but for a longer time during biodegradation. In the case of the PHB blended with limonene, a decrease in the mechanical properties of about 30% was marked at the beginning of the biodegradation. After this decrease, the material seemed to be relatively stable during the tested period. These additives are fully biobased and do not prohibit the ultimate biodegradation of the material at the end of its service life.

## Figures and Tables

**Figure 1 ijms-23-14079-f001:**
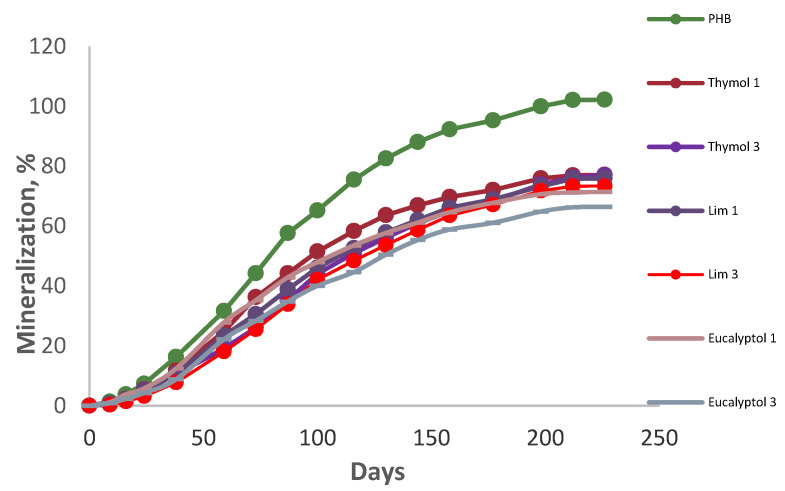
Mineralization of the additives at room temperature.

**Figure 2 ijms-23-14079-f002:**
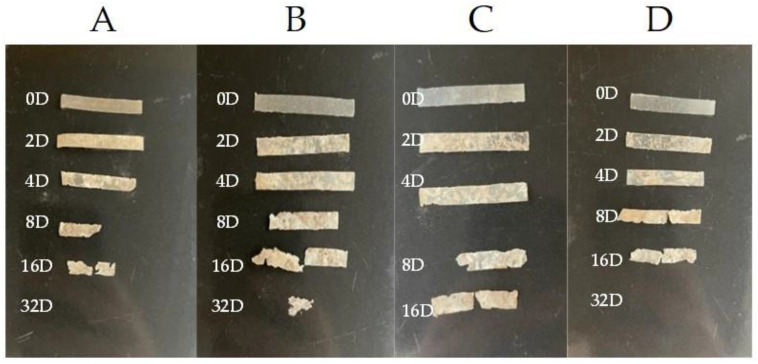
Physical shapes of the samples during incubation, observed at different times: 0, 2, 4, 8, 16, and 32 days: (**A**) pure PHB, (**B**) PHB + 3% eucalyptol, (**C**) PHB + 3% limonene, and (**D**) PHB + 3% thymol.

**Figure 3 ijms-23-14079-f003:**
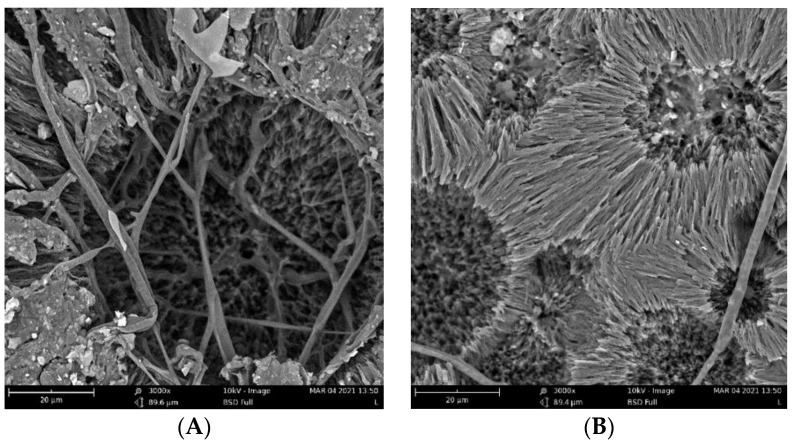
SEM images of the samples after 16 days of incubation: (**A**) pure PHB, (**B**) limonene 3%, (**C**) thymol 3%, and (**D**) eucalyptol 3%.

**Figure 4 ijms-23-14079-f004:**
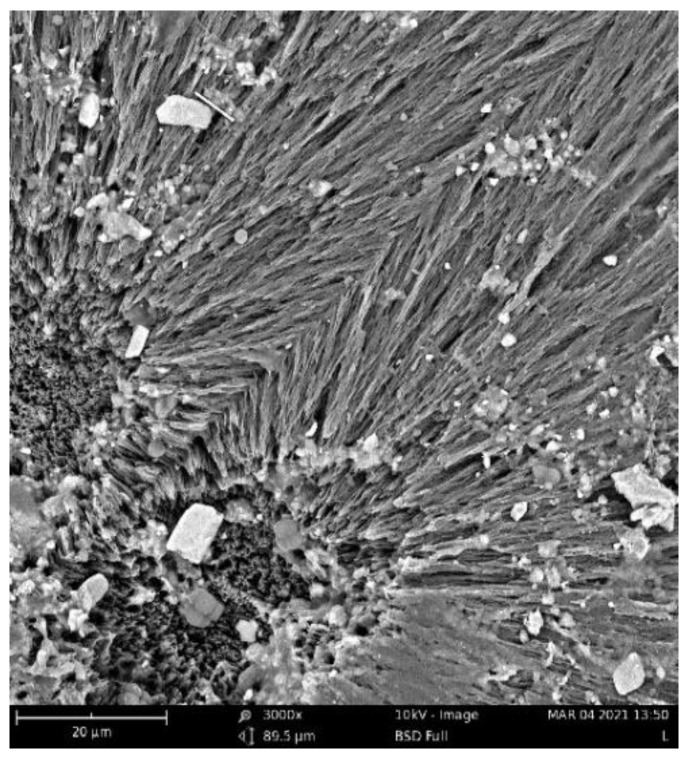
SEM image of PHB/eucalyptol 3% after 32 days of incubation.

**Figure 5 ijms-23-14079-f005:**
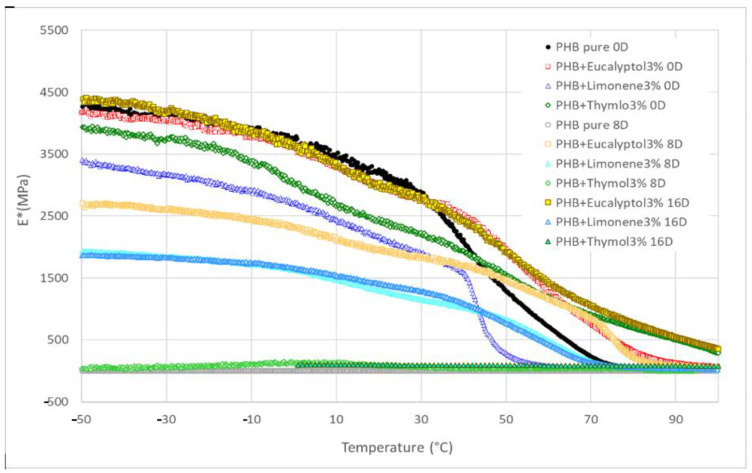
DMA: Complex modulus E* for PHB mixtures before and after degradation.

**Figure 6 ijms-23-14079-f006:**
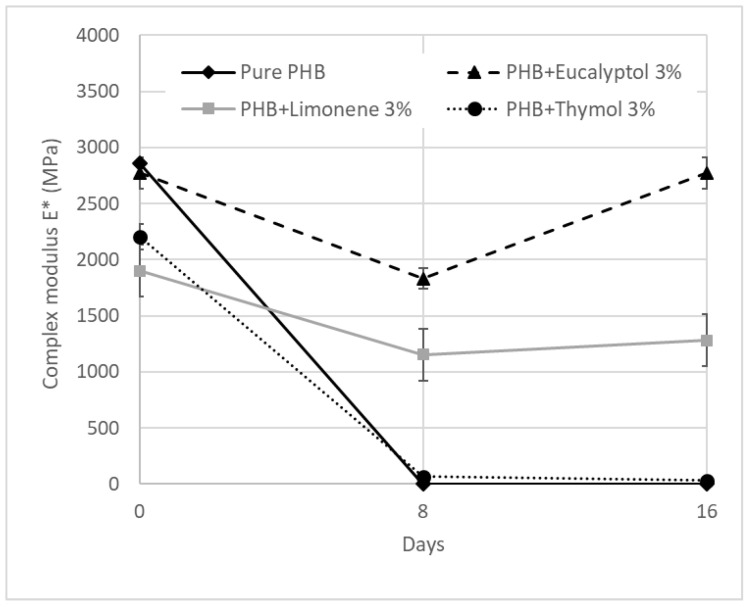
DMA: Complex modulus E* for PHB mixtures at 30 °C.

**Table 1 ijms-23-14079-t001:** DSC: Crystallinity (second heating).

Sample	Crystallinity (%)				
	0 Days	4 Days	8 Days	16 Days	32 Days
Pure PHB	69	67	61	63	-
Eucalyptol 3%	70	64	62	62	66
Limonene 3%	71	66	68	67	66
Thymol 3%	66	67	66	65	-

**Table 2 ijms-23-14079-t002:** DMA: Complex modulus E* for PHB mixtures at 30 °C.

Sample	Complex Modulus E* (MPa) at 30 °C		
	0 Days	8 Days	16 Days
Pure PHB	2862	0	-
Eucalyptol 3%	2774	1832	2774
Limonene 3%	1901	1155	1282
Thymol 3%	2205	67	31

**Table 3 ijms-23-14079-t003:** DSC results of PHB/terpenoid blends (second heating).

Sample	Incubation Time (Day)	T_g_ (°C)	T_c_ (°C)	T_cc_ (°C)	T_m_ (°C)	
					T_m1_	T_m2_
Pure PHB	0	5.66	82.85	94.25	-	172.00
	16	5.80	83.06	94.23	168.23	173.58
	32	-	-	-	-	-
PHB + Eucalyptol 3%	0	4.94	84.37	93.40	167.73	172.89
	16	5.50	87.26	95.07	167.42	172.77
	32	5.67	85.10	97.56	166.20	172.93
PHB + Limonene 3%	0	4.05	77.33	94.56	166.79	172.86
	16	4.73	85.07	94.23	166.11	172.56
	32	4.84	88.28	95.08	166.63	172.16
PHB + Thymol 3%	0	4.65	69.49	87.75	167.14	172.75
	16	4.87	86.77	94.91	166.90	172.11
	32	-	-	-	-	-

**Table 4 ijms-23-14079-t004:** The properties of essential oils blended with PHB.

Additive (Purity)	Molecular Weight	Made in	Other Properties: Melting Point (MP), Bubble Point (BP)
Camphor (98%)	152.23	USA	MP = 179–181 °C
Thymol (98.5%)	150.22	India	PB = 232 °C, MP = 48–51 °C
Carvacrol (≥98%)	150.22	India	MP = 3–4 °C, BP = 236–237 °C
Trans-anethole (99%)	148.20	Spain	MP = 20–21 °C, BP = 234–237 °C
Eugenol (99%)	164.2	Germany	MP = 12–10 °C, BP = 254 °C
Eucalyptol (99%)	154.25	USA	MP = 1–2 °C, BP = 176–−177 °C
Carvone (98%)	150.22	China	BP = 227–230 °C
Limonene (96%)	136.23	Spain	BP = 175–177 °C
Trans-cinnamaldehyde (99%)	132.16	China	MP = −9–−4 °C, BP = 250–252 °C
Nerol	154.25	USA	BP = 103–105 °C

## Data Availability

Not applicable.

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
