# Peer review of "Selected Simple Natural Antimicrobial Terpenoids as Additives to Control Biodegradation of Polyhydroxy Butyrate"

_ijms, 2022, doi:10.3390/ijms232214079_

Round 1
Reviewer 1 Report
The paper discusses the addition of terpenoids to PHB in order to control their biodegradation. This is a very interesting and important topic as PHB seems to be a very promising polymer for biodegradation in different environments and a very promising polymer to decrease the environmental impact of polymer materials. Nowadays, one of the main challenges is to modulate the (bio)degradation rate of PHB and this paper deal with this important issue and demonstrates interesting results. Different kinds of terpenoids were blended with PHB, it seems that Eucalyptol was evaluated as the most effective biobased additive to retard the biodegradation of PHB. The paper deserves to be published after some minor revisions:
-figure 2 shows samples through incubation at 6 different times but the figure shows only 5 samples (please indicate which incubation times are not shown, 0 days ?)
-the crystallinity rate calculation formula was not indicated in the material and methods. Please indicate it. Even though the rate of additives is quite low in PHB (1, 3%) but are the percentage of the additives been taken into consideration to calculate the crystallinity rate?
- Table 2 and figure 5 represent the same results may be only table 2 it’s enough to demonstrate the results
-Too much information in figure 6, not easy to follow all the curves, maybe it can be separated depending on the incubation times. Please correct the legend in figure 6 (complex and not complex)
-what’s the auteur thinks about the process temperature (185°C) and the impact during blending?
Reviewer 2 Report
Please find my comments in the attached file.

Round 2
Reviewer 2 Report
After revision, I suggest to accept the manuscript in the present form.